# Bioactive Compounds, Nutritional Value, and Potential Health Benefits of Indigenous Durian (*Durio Zibethinus* Murr.): A Review

**DOI:** 10.3390/foods8030096

**Published:** 2019-03-13

**Authors:** Nur Atirah A Aziz, Abbe Maleyki Mhd Jalil

**Affiliations:** School of Nutrition and Dietetics, Faculty of Health Sciences, Universiti Sultan Zainal Abidin, Kuala Nerus 21300, Malaysia; atirah_aziz@ymail.com

**Keywords:** durian, esters, thioacetals, thioesters, volatile compounds, polyphenols, propionate

## Abstract

Durian (*Durio zibethinus* Murr.) is an energy-dense seasonal tropical fruit grown in Southeast Asia. It is one of the most expensive fruits in the region. It has a creamy texture and a sweet-bitter taste. The unique durian flavour is attributable to the presence of fat, sugar, and volatile compounds such as esters and sulphur-containing compounds such as thioacetals, thioesters, and thiolanes, as well as alcohols. This review shows that durian is also rich in flavonoids (i.e., flavanols, anthocyanins), ascorbic acid, and carotenoids. However, limited studies exist regarding the variation in bioactive and volatile components of different durian varieties from Malaysia, Thailand, and Indonesia. Experimental animal models have shown that durian beneficially reduces blood glucose and cholesterol levels. Durian extract possesses anti-proliferative and probiotics effects in in vitro models. These effects warrant further investigation in human interventional studies for the development of functional food.

## 1. Introduction

*Durio zibethinus* Murr. (family *Bombacaceae*, genus *Durio*) is a seasonal tropical fruit grown in Southeast Asian countries such as Malaysia, Thailand, Indonesia, and the Philippines. There are nine edible *Durio* species, namely, *D. lowianus, D. graveolens* Becc., *D. kutejensis* Becc., *D. oxleyanus* Griff., *D. testudinarum* Becc., *D. grandiflorus* (Mast.) Kosterm. ET Soeg*., D. dulcis* Becc., *Durio* sp., and also *D. zibethinus* [1]. However, only *Durio zibethinus* species have been extensively grown and harvested [2]. In Malaysia, a few varieties have been recommended for commercial planting such as D24 (local name: *Bukit Merah*), D99 (local name: *Kop Kecil*), and D145 (local name: *Beserah*). In Thailand, durian species were registered based on local names such as *Monthong, Kradum*, and *Puang Manee*. There are similar varieties between Malaysian and Thailand but with different name as follows: D123 and *Chanee*, D158 and *Kan Yao*, and D169 and *Monthong* [3]. Similar to Thailand, durian varieties in Indonesia are registered based on their local names, such as *Pelangi Atururi, Salisun, Nangan, Matahari*, and *Sitokong* [1,4].

The durian fruit shape varies from globose, ovoid, obovoid, or oblong with pericarp colour ranging from green to brownish [1] (Figure 1). The colour of edible aril varies from one variety to the others and fall in between the following: yellow, white, golden-yellow or red [5]. It is eaten raw and has a short shelf-life, from two to five days [5,6]. Fully ripened durian fruit has a unique taste and aroma, and is dubbed “king of fruits” in Malaysia, Thailand, and Singapore. The unique taste and aroma is attributed to the presence of volatile compounds (esters, aldehydes, sulphurs, alcohols, and ketones) [6,7].

Hundreds of volatile compounds have been identified in Malaysian, Thailand, and Indonesian durian varieties such as esters (ethyl propanoate, methyl-2-methylbutanoate, propyl propanoate), sulphur compounds (diethyl disulphide, diethyl trisulphide and ethanethiol), thioacetals (1-(methylthio)-propane), thioesters (1-(methylthio)-ethane), thiolanes (3,5-dimethyl-1,2,4-trithiolane isomers), and alcohol (ethanol) [6,7]. However, the bioactivity of these compounds has not yet been thoroughly explored. A study by Alhabeeb et al. (2014) showed that 10 g/day inulin propionate ester (a synthetic propionate) releases large amounts of propionate in the colon. This subsequently increases perceived satiety (increased satiety and fullness, decreased desire to eat) [8]. Chambers et al. (2015) showed that the same propionate ester (400 mmol/L) increased peptide YY (PYY) and glucagon-like peptide 1 (GLP-1) in primary cultured human colonic cells. This study also showed that 10 g/day of inulin-propionate ester reduced energy intake (14%) compared with the control (inulin) [9].

Durian is also rich in polyphenols such as flavonoids (flavanones, flavonols, flavones, flavanols, anthocyanins), phenolic acids (cinnamic acid and hydroxybenzoic acid), tannins, and other bioactive components such as carotenoids and ascorbic acid [10,11,12,13,14,15,16,17,18,19,20,21,22,23,24,25]. Current epidemiological studies have suggested that polyphenols decrease the risk of chronic diseases (e.g., cardiovascular diseases, cancers and diabetes) [26,27,28,29,30]. However, polyphenols might act synergistically with other phytochemicals [26]. However, currently, there are limited studies exploring the health benefits of bioactive components in durian. Hence, we aimed to review the nutritional and bioactive compounds present in durian varieties from Thailand, Indonesia, and Malaysia, as well as to explore the potential health benefits of durian. 

## 2. Nutritional Composition of Different Durian Varieties

The energy content of durian is in the range of 84–185 kcal per 100 g fresh weight (FW) (Table 1) [6,18,19]. This range is somewhat similar to that of the United States Department of Agriculture (USDA), Malaysian, and Indonesian food composition databases [20,21,22]. Durian aril of the Thailand variety of *Kradum* showed the highest energy content at 185 kcal compared with other durian varieties [6,12,13]. Indonesian variety of *Hejo* showed the lowest energy content at 84 kcal per 100 g FW of durian aril [6]. The higher and lower energy contents are attributed to the difference in carbohydrate content. The carbohydrate content varies between different durian varieties in the range between 15.65 to 34.65 g per 100 g FW [6,12,13]. The range of carbohydrate content is similar to that of USDA, Malaysian and Indonesian food composition data, at 27.09 g, 27.90 g, and 28.00 g per 100 g FW, respectively [31,32,33]. The energy content of durian is the highest compared with other tropical fruits such as mango, jackfruit, papaya, and pineapple [31].

Protein content of different durian varieties is in the range of 1.40 to 3.50 g per 100 g FW [6,12,13]. This range is similar to that of USDA, Malaysian, and Indonesian food composition data, at 1.47 g, 2.70 g, and 2.50 g per 100 g fresh weight (FW), respectively [31,32,33]. Durian contains a high amount of fat and is in the range of 1.59 to 5.39 g per 100 g FW, a figure comparable to the data from USDA, Malaysian, and Indonesian food composition databases at 5.33 g, 3.40 g, and 3.00 g of fat per 100 g FW, respectively [6,12,13,31,32,33]. The fat content of durian is somewhat comparable to one-third of ripe olives [31]. Total sugar of Malaysian, Thailand, and Indonesian durian varieties is in the range of 7.52 to 16.90 g, 14.83 to 19.97 g, and 3.10 to 14.05 g per 100 g FW, respectively (Table 2). The Thailand variety of *Kradum* showed the highest total sugar, at 19.97 g per 100 g FW. Sucrose was the predominant sugar in durian, with 5.57 to 17.89 g per 100 FW, followed by glucose, fructose, and maltose. However, the Malaysian variety of D24 contains higher amounts of fructose than glucose.

Table 3 shows fatty acid compositions of different durian varieties. Thailand durian varieties showed higher monounsaturated fatty acids (MUFA) than saturated fatty acids (SFA) and polyunsaturated fatty acids (PUFA), with exception of *Monthong*. Palmitic acid (16:0) was the major SFA, in the range of 84.57 to 1696.00 mg per 100 g FW, while oleic acid (18:1) was the major MUFA found in the matured or fully ripened durian (64.89 to 2343.30 mg per 100 g FW). However, each study used a different technique for fatty acid analysis. Gas chromatography was used by Charoenkiatkul et al. (2015) while high pressure liquid chromatography was used by Haruenkit et al. (2010) [13,14]. Both MUFA and SFA might be involved in various metabolic pathways, including the regulation of transcription factors and the expression of multiple genes related to inflammatory processes [37,38,39].

Table 4 shows the mineral compositions of ripe Thailand durian. Durian is high in potassium in the range from 70.00 to 601.00 mg per 100 g FW [11,13,14,31,32,33]. This is comparable to potassium-rich fruit such as banana, with the value of 358.00 mg per 100 g FW [31]. Phosphorus, magnesium, and sodium are in the range of 25.79 to 44.00, 19.28 to 30.00, and 1.00 to 40.00 mg per 100 g FW, respectively. Durian is also a source of iron, copper, and zinc with the range of 0.18 to 1.90, 0.12 to 0.27 and 0.15 to 0.45 mg per 100 g FW, respectively. The Thailand variety of *Chanee* showed the highest level of iron, zinc and potassium among the studied durian [12,19,20,21,22,29]. Durian also contains vitamin A, different types of vitamin B, and vitamin E [13,14,15,31,32,33].

Table 5 shows soluble, insoluble, and total dietary fibres in Thailand durian varieties. However, there are limited data available for Indonesian and Malaysian varieties. The total dietary fibre is in the range from 1.20 to 3.39 g per 100 g FW for Thailand *Monthong* variety. However, it must be noted that different analyses were used between studies. Soluble dietary fibre varied from 0.74 g (*Puang Manee)* to 1.40 g (*Monthong*) per 100 g FW while insoluble dietary fibre is in the range from 0.60 g (*Kan Yao)* to 2.44 g (*Chanee*) per 100 g FW [10,12,16].

## 3. Bioactive Compounds and Antioxidant Capacity

Total polyphenols content of ripe durian is in the range of 21.44 to 374.30 mg gallic acid equivalent (GAE) per 100 fresh weight (FW) (Table 6). The Thailand variety of *Monthong* showed the highest polyphenols content with 374.30 mg GAE per 100 FW compared with other durian varieties [10,11,12,13,14,17,18,19,20,21]. Total flavonoid content of different durian varieties is in the range of 1.90 to 93.90 mg catechin equivalent (CE) per 100 g FW [10,11,12,14,16,17,18,19,20,21,22]. This review found three main flavonoids, namely flavanones (hesperetin and hesperidin), flavonols (morin, quercetin, rutin, kaempferol, myricetin), and flavones (luteolin and apigenin). Hesperetin was quantified in Thailand durian variety in the range of 260.99 to 1110.23 μg per 100 g FW [16]. The predominant flavonol was quantified in *Monthong* as quercetin with 2549.30 mg per 100 g FW [18,19,20]. Morin, a type of flavonol, was also detected in mature and ripe durian variety of *Monthong* in the range from 110.00 to 550.00 μg per 100 g FW [19]. Rutin and kaempferol were quantified in the range of 163.90 to 912.05 μg per 100 g FW and 131.64 to 2200.00 μg per 100 g FW, respectively [18]. Lowest and highest myricetin contents were quantified in *Kradum* and *Monthong*, at 320.00 μg and 2159.27 μg per 100 g FW, respectively [19,23]. The main flavones were identified in durian as luteolin and apigenin in the range of 279.29 to 509.09 µg and 509.09 to 791.94 µg per 100 g FW, respectively [16,18,19,23]. The total flavanol content is in the range of 0.13 mg to 5.18 mg CE per 100 g FW [11,12,14,17,18,19,20,21]. The anthocyanins content is in the range 0.32 to 633.44 mg cyanidin-3-glucoside equivalent (CGE) per 100 g FW [18,19,22].

Phenolic acids in durians belong to hydroxycinnamic acid (caffeic, *p*-coumaric, ferulic, *p*-anisic acid) and hydroxybenzoic acid (gallic and vanillic acid) derivatives. Cinnamic acid, caffeic acid, *p*-coumaric acid, and *p*-anisic acid were quantified in *Monthong* variety in the range of 600.00 to 660.00 μg, 31.08 to 490.00 μg, 29.22 to 600.00 μg, and 1.48 μg per 100 g FW, respectively [19,21]. Ferulic acid was identified in *Chanee, Puang Manee*, and *Monthong* in the range of 215.95 μg, 158.67 μg and 414.40 μg per 100 g FW, respectively [18,21]. Gallic acid is the main hydroxybenzoic acid identified in *Chanee*, *Monthong*, and *Puang Manee*, at 1416.00, 2072.00, and 4760.10 μg per 100 g FW respectively [18].

Total carotenoids content was higher in Thailand compared with Malaysian variety in the range of 222.88 µg to 6000.00 µg and 5.13 µg to 8.22 µg BCE per 100 g FW, respectively [11,17,24]. Thailand durian varieties contain minor amount of β-carotene, α-carotene, β-cryptoxanthin, lycopene, lutein, and zeaxanthin [13,18,24,25]. Carotenoid content varies in durian and depending on factors such as variety, part of the plant, degree of maturity, climate, soil type, growing conditions and geographical area of production [40]. Tannins have been identified in *Monthong* variety in the range from 29.60 to 296.00 μg per 100 g FW [11,14,21,22]. Ascorbic acid content in the Malaysian variety is in the range from 1.93 to 8.62 mg per 100 g FW [17]. The Thailand variety of *Monthong* variety showed the highest ascorbic acid, with 347.80 mg per 100 g FW [14].

Durian is rich in bioactive polyphenols and hence, exerts antioxidant potential. Table 7 shows antioxidant capacity of durians based on 1-diphenyl-2-picrylhydrazyl radical (DPPH), ferric ion reducing antioxidant power (FRAP), oxygen radical absorbance capacity (ORAC), cupric reducing antioxidant capacity (CUPRAC), hydrophilic oxygen radical absorbance capacity (H-ORAC), and 2,2′-azino-bis-3-ethylbenzthiazoline-6-sulphonic acid (ABTS) assays [12,13,14,15,18,22,23,40,41]. Antioxidant activities of Thailand durian varieties were in the range from 97.93 to 1366.16 μM Trolox equivalents (TE) per 100 g FW for DPPH assay, 71.84 to 749.08 μM TE per 100 g FW for FRAP assay, 1903.40 to 2793.90 μM TE per 100 g FW for ORAC assay, 427.65 to 1075.60 μM TE per 100 g FW for CUPRAC assay, and from 265.86 to 2352.70 μM TE per 100 g FW for ABTS assay [12,13,14,18,41,42]. Antioxidant activity of the unknown Malaysian durian variety was 1838.00 μM TE per 100 g FW, as determined using H-ORAC assay [15]. Antioxidant activity of unknown durian (Chinese study) was 498.00 μM TE per 100 g FW as assayed using ABTS [23].

## 4. Volatile Components

Durian is rich in volatiles esters, alcohols, ketones and sulphur (Table 8). These volatile compounds gave durian a unique flavour and taste. Chin et al. (2007) reported 39 volatile compounds in the three Malaysian durian varieties, D2, D24 and D101 [7]. A total of 44 volatile compounds were identified in Indonesian durian varieties of *Ajimah, Hejo, Matahari*, and *Sukarno* [42]. The main volatile component in durian is sulphur. Ethanethiol, propanethiol, diethyl disulphide, ethyl propyl disulphide, ethyl propyl disulphide, and diethyl trisulphide were the predominant sulphur compounds identified in Malaysian durian variety. The sulphur compounds in Malaysian varieties were 97% higher than Indonesian variety.

The volatile sulphur compounds (VSCs) have a smell resembling onion [43]. Durians from Indonesia have lower VSCs and contributed to the less sulphuric odour in *Hejo* and *Sukarno. Sukarno* has sweet odour, while *Hejo* has the mildest sulphuric odour among the studied durians varieties in Indonesia [6]. There were an additional 12 VSCs identified in Indonesian variety of *Cane*, *Kodak*, and *Bobo* [44]. The VSCs were identified as *S*-ethyl thioacetate, 1-hydroxy-2-methylthioethane, methyl 2- methylthioacetate, dimethyl sulfone, *S*-ethyl thiobutyrate, ethyl 2-(methylthio) acetate, 2-isopropyl-4-methylthiazole, *S*-isopropyl 3-(methylthio), *S*-methyl thiohexanoate, 5-methyl-4-mercapto-2-hexanone benzothiazole, 3,4-dithia-2-ethylthiohexane, and *S*-methyl thiooctanoate, 3,5- dimethyltetrathiane.

Ethanol was the predominant alcohol compound in Malaysian and Indonesian varieties in the range from 590.00 to 720.00 ng per g and 419.90 to 1091.30 ng per g fresh weight (FW), respectively. Another three alcohols were detected in durian as 2-methyl-1-butanol, 3-methyl-1-butanol and 2,3-butanediol in Malaysian and Indonesian durian [7,45]. Weenen et al. (1996) detected additional alcohols in durian *Cane, Kodak*, and *Bobo* from Indonesia as hexadecanol, 9-octadecen-1-ol (*cis* and *trans*), and isobutyl alcohol [44]. Voon et al. (2007) and Chin et al. (2008) detected 1-propanol, 1-butanol and 1-hexanal in Malaysian variety of *Chuk*, D101, D2, MDUR78, and D24 [45,46].

3-Hydroxy-2-butanone (a ketone) was identified in the Indonesian variety in the range of 56.20 to 84.20 ng per g FW [7,42]. Durian *Matahari* showed the highest amount of 3-hydroxy-2-butanone with 84.20 ng per g FW and the lowest in *Hejo* with 56.20 ng per g FW [42]. 3-Hydroxy-2-butanone (common name: acetoin) has a pleasant yogurt creamy odour and a fatty butter taste. It is present in vinegar and alcoholic beverages [47]. Another ketone was identified as 2-hydroxy-3-pentanone in durian varieties from Indonesia, *Kodak, Cane*, and *Bobo* [44]. However, 2-hydroxy-3-pentanone has an unfavourable odour like fishy and earthy [48]. For aldehydes, acetaldehyde detected in Indonesian variety of *Hejo, Matahari*, and *Ajimah* in the range of 33.90 to 62.20 ng per g FW [42]. Acetaldehyde contributed to the fruity and sweet aroma in durian [49].

Esters are the second most abundant bioactive compounds in durian after sulphur. Esters were the volatiles that contributed to the sweet odour to durian, more than aldehyde, while aldehyde contributed more to the fruity note. The major ester compounds in Malaysian durian varieties (D101, D2, and D24) were characterized as propyl 2-methylbutanoate, ethyl propanoate, propyl propanoate and methyl 2-methylbutanoate [7]. Ethyl 2-methylbutanoate was detected as major ester in Indonesian varieties of *Hejo, Matahari*, and *Ajimah* [42]. Ethyl propanoate, methyl propanoate, propyl propanoate, ethyl 2-methyl propanoate and 3-methylbutyl propanoate volatiles have similar structure and differ only in the number of carbon atoms (ethyl, methyl) [50]. Ethyl propanoate is the major ester detected in Malaysian and Indonesian durians in the range of 0.00 to 3110.00 ng per g FW. It was noted that Malaysian durian variety showed much higher content of ethyl propanoate than Indonesian variety [7,42].

## 5. Health Benefits of Durian

Durian is rich in macronutrients (sugars and fat) and micronutrients (potassium), dietary fibres, and bioactive and volatile compounds. An intake of one serving size of durian aril (155 g) contributes to 130 to 253 kcal and is equivalent to one large pear and four small apples without skin, respectively [6,31,32,33]. Durian is energy-dense due to sugar and fat content and hence, might contribute to daily energy intake and will also increase postprandial blood glucose.

### 5.1. Effects of Durian on Blood Glucose

Durian is high in sugar, but supplementation of 5% freeze-dried *Monthong* (Thailand variety) in 1% cholesterol-enriched diets in rats for 30 days did not raise the plasma glucose level compared with control diet [41]. In humans, Robert et al. (2008) showed that durian had the lowest glycaemic index (GI = 49) compared with watermelon (GI = 55), papaya (GI = 58), and pineapple (GI = 90) [51]. The low GI value for durian might be due to the presence of fibre and fat. Fibre slows digestion in the digestive tract and will slow down the conversion of the carbohydrate to glucose, thus lower the GI of food [52]. Fat does not have a direct effect on blood glucose response, but it may influence glycaemic response indirectly by delaying gastric emptying, and thus slowing the rate of glucose absorption [53].

Durian is rich in potassium and is similar to potassium-rich fruit, i.e., banana [31]. A meta-analysis study showed that there was a linear dose-response between low serum potassium and risk of type 2 diabetes mellitus [54]. Chatterjee et al. (2017) demonstrated that potassium chloride supplementation reduced the worsening effect of fasting glucose in African-Americans compared with placebo [55]. Collectively, the evidence has shown that potassium content in durian might play a role in the regulation of blood glucose. The effect of durian on blood glucose has not been thoroughly explored both in animal and human studies, and hence, warrants further investigation. Potassium might play a role in glucose homeostasis but might also have negative implications in certain conditions. For instance, those with chronic kidney disease (CKD), diabetes mellitus (DM), and heart failure (HF) or on pharmacological therapies may develop hyperkalaemia [56].

### 5.2. Cholesterol-Lowering Properties of Durian

Anti-atherosclerotic properties of durian aril have been reported in experimental rat models [10,11,20,22,40,41]. Previous in vitro and in vivo studies investigated the health benefits of durian (*Monthong* variety) on lipid profiles [10,11,22]. Haruenkit et al. (2007) showed that rats fed with durian significantly (*p* < 0.05) reduced postprandial plasma total cholesterol (TC) and low-density lipoprotein cholesterol (LDL-C) with 14.9% and 21.6%, respectively, compared with control group [10]. Gorinstein et al. (2011) showed a reduction in the levels of plasma TC (12.1%), LDL-C (13.3%), and triglycerides (TG) (14.1%) compared with the control group [11]. The results were consistent when tested with other durian from Thailand varieties (*Chanee* and *Kan Yao*) compared with control. Leontowicz et al. (2011) showed that rats supplemented with ripe durian had significantly lowered TG (26.3%), but not significant in TC (4.8%) and LDL-C (6.3%). Histological analysis demonstrated that ripe durian protected the liver and aorta from exogenous cholesterol loading and protected the intimal surface area of the aorta [20]. Durian also demonstrated the ability to hinder postprandial plasma lipids compared with snake fruit and mangosteen [10,11,22]. Previous studies have showed that propionate (0.6 mmol/L) inhibited fatty acid and cholesterol synthesis in isolated rat hepatocytes [57]. In our review, three different propionate esters were identified, i.e., ethyl propionate, methyl propionate and propyl propionate. These esters could be a potent inhibitor for free fatty acids and cholesterol synthesis but this warrants further investigations. However, these esters are highly volatile and could be easily vaporized during sample processing and storage [57].

### 5.3. Anti-Proliferative Activity

The polyphenol and flavonoid contents of durian are in the range of 21.44 to 374.30 mg GAE and 1.90 to 93.90 mg CE per 100 g FW. The mechanisms of action of polyphenols strongly relates to their antioxidant activity. Polyphenols are known to decrease the level of reactive oxygen species in the human body [58]. The phenolic groups present in the polyphenol structure can accept an electron to form relatively stable phenoxyl radicals, thereby disrupting chain oxidation reactions in cellular components [59]. On the other hand, polyphenols could induce apoptosis and inhibit cancer growth [60,61,62,63]. There are many studies pointing out an essential role of polyphenolic compounds as derived from vegetables, fruits, or herbs in the regulation of epigenetic modifications, resulting in the antiproliferative protection [64]. Jayakumar and Kanthimathi studied the anti-proliferative activity of durian using a breast cancer cell line (MCF-7). This study showed that durian fruit can be considered as potential sources of polyphenols with protective effects against nitric oxide-induced proliferation of MCF-7 cells, an oestrogen receptor-positive human breast cancer cell line [65]. At a concentration of 600 µg/mL, durian fruit extracts inhibited MCF-7 cell growth by 40%. However, an in vivo study is needed to confirm this effect.

### 5.4. Probiotic Effects

Durian aril is rich in sugar with total sugar content between 3.10 to 19.97 g per 100 g FW. The moisture content of durian aril is 56.1 g to 69.3 g per 100 g FW and pH between 6.9 to 7.6 [5,13,31]. These could be an optimum condition for bacteria fermentation. Durian aril is fermented after being left at room temperature for a few days and turns sour and watery. In Malaysia, underutilised durian aril is fermented (spontaneous and uncontrolled) to a product known as *Tempoyak* [66]. *Tempoyak* is widely used as seasoning in cooking. According to Leisner et al. (2001) lactic acid bacteria (LAB) are the predominant microorganisms in *Tempoyak* [67]. The LAB microorganisms were identified as *Lactobacillus plantarum*. However, other species including *Lactobacillus fersantum, Lactobacillus corynebacterium, Lactobacillus brevis, Lactobacillus mali, Lactobacillus fermentum, Lactobacillus durianis, Lactobacillus casei, Lactobacillus collinoides, Lactobacillus paracasei* and *Lactobacillus fructivorans* were also reported in *Tempoyak* [67,68,69,70]. Khalil et al. (2018) and Ahmad et al. (2018) recently demonstrated the potential of *Tempoyak* as a source of probiotics. The study by Khalil et al. (2018) isolated seven *Lactobacillus* strains that belonged to five different species of the genus *Lactobacillus*, including one *Lactobacillus fermentum* (DUR18), three *Lactobacillus plantarum* (DUR2, DUR5, DUR8), one *Lactobacillus reutri* (DUR12), one *Lactobacillus crispatus* (DUR4), and one *Lactobacillus pentosus* (DUR20) from *Tempoyak*. These strains were able to produce exopolysaccharide (EPS) and had great potency to withstand the extreme conditions, either at low pH 3.0, in 0.3% bile salts or in in vitro model of gastrointestinal conditions [69]. EPS has the prebiotic potential to positively affect the gastrointestinal (GIT) microbiome and may reduce cholesterol [70]. Ahmad et al. (2018) isolated *Lactobacillus plantarum* from *Tempoyak* and showed good probiotic properties including acid and bile salt tolerance, antioxidative, antiproliferative effects, and remarkable adhesion on colon adenocarcinoma cell line (HT-29 cell lines) [71].

## 6. Conclusions

Durian is rich in macronutrients (sugars and fat) and micronutrients (potassium), dietary fibre, and volatile compounds. Durian is an energy-dense fruit due to high sugar and fat content and, hence, might contribute to daily energy intake and increase postprandial blood glucose. Durian is also rich in bioactive polyphenols and hence possessed strong in vitro antioxidant capacity. However, the bioactivity of these polyphenols in animal or human studies is still scarce and needs further investigation. The major volatile compounds have been identified in the Malaysian, Thai, and Indonesian durian varieties as esters (ethyl propanoate, methyl-2- methylbutanoate, propyl propanoate), sulphurs (diethyl disulphide, diethyl trisulphide and ethanethiol), thioacetals (1-(methylthio)-propane), thioesters (1-(methylthio)-ethane), thiolanes (3,5-dimethyl-1,2,4-trithiolane isomers), and alcohol (ethanol). Both in vitro and in vivo animal studies showed that durian possessed anti-hyperglycaemic, anti-atherosclerotic, anti-proliferative, and probiotic effects. Durian is rich in bioactive compounds, and hence can be used as an active ingredient for the development of functional foods. Further human interventional studies are warranted to explore the health benefits of functional foods prepared with durian.

## Figures and Tables

**Figure 1 foods-08-00096-f001:**
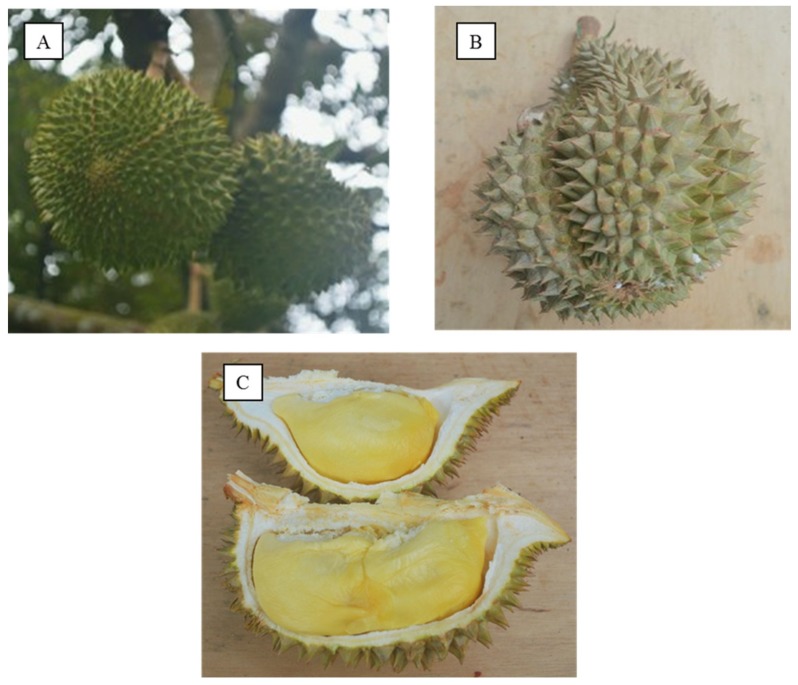
(**A**) Durian tree with fruit. (**B**) Durian fruit with its spiny rind. (**C**) Durian aril (flesh).

**Table 1 foods-08-00096-t001:** Nutritional composition of durian aril (flesh) of different durian varieties (g per 100 g fresh weight).

Durian Variety	Indonesian Variety				Thailand Variety				Unknown Variety [31]	Unknown Variety [32]	Unknown Variety [33]
	*Ajimah*	*Hejo*	*Matahari*	*Sukarno*	*Monthong*	*Chanee*	*Kradum*	*Kobtakam*			
Nutrients											
Energy (kcal) [6] * [31,32,33]	151	84	163	134	134–162	145	185	145	147	153	134
Carbohydrate (g) [6] * [12,13,31,32,33]	28.90	15.65	34.65	27.30	21.70–27.10	20.13	29.15	21.15	27.09	27.90	28.00
Protein (g) [6] * [12,13,31,32,33]	2.36	1.76	2.33	2.13	1.40–2.33	3.10	3.50	2.86	1.47	2.70	2.50
Fat (g) [6] * [12,13,31,32,33]	2.92	1.59	1.69	1.86	3.10–5.39	4.48	4.67	4.40	5.33	3.40	3.00

* For [6], energy was calculated by Atwater factor (1 g protein = 4 kcal, 1 g carbohydrate = 4 kcal, 1 g fat = 9 kcal) [34].

**Table 2 foods-08-00096-t002:** Sugar composition of different durian varieties (g per 100 g fresh weight).

Sugars	Fructose [13,35,36]	Glucose [13,35,36]	Sucrose [13,35,36]	Maltose [13,35]	Total Sugar [6] * [13,35,36]
Malaysian Variety					
*Durian Kampung*	1.60	2.21	12.58	0.51	16.90
D2	1.66	2.51	7.70	NA	11.87
D24	0.76	0.73	6.03	NA	7.52
MDUR78	1.82	2.77	8.02	NA	12.61
D101	1.29	1.97	5.57	NA	8.83
*Chuk*	1.28	1.87	10.65	NA	13.80
Thailand Variety					
*Monthong*	0.15	0.74	13.69	0.25	14.83
*Chanee*	0.26	0.58	15.71	0.00	16.55
*Kradum*	0.33	0.71	17.89	1.04	19.97
*Kobtakam*	0.10	0.45	17.30	0.26	18.11
Indonesian Variety					
*Ajimah*	NA	NA	NA	NA	14.05
*Hejo*	NA	NA	NA	NA	3.10
*Matahari*	NA	NA	NA	NA	8.14
*Sukarno*	NA	NA	NA	NA	8.12

* Total sugar is the sum of each individual sugar except for [6], NA, not available.

**Table 3 foods-08-00096-t003:** Fatty acid (FA) composition of different durian varieties (mg per 100 g fresh weight).

Thailand Variety		*Monthong*	*Chanee*	*Kradum*	*Kobtakam*
Fatty Acid Name	Nomenclature	Fatty Acids Composition			
Decanoic (Capric) [14]	C 10:0	0.11–0.19	NA	NA	NA
Dodecanoic (Lauric) [13]	C 12:0	3.07	16.00	16.68	9.63
Tetradecanoic (Myristic) [13,14]	C 14:0	1.50–30.70	64.00	41.70	32.10
Hexadecanoic (Palmitic) [13,14]	C 16:0	84.57–1473.60	1696.00	1626.30	1508.70
*cis*-9-Hexadecenoic (Palmitoleic) [13]	C 16:1	122.80	192.00	125.10	160.50
Octadecanoic (Stearic) [13,14]	C 18:0	3.48–61.40	64.00	83.40	96.30
*cis*-9-Octadecenoic (Oleic) [13,14]	C 18:1 *n*-9	64.89–1074.50	1952.00	2376.90	2343.30
*cis*-9,12-Octadecadienoic (Linoleic) [13,14]	C 18:2 *n*-6	10.78–184.20	128.00	125.10	160.50
*cis*-6,9,12-Octadecatrienoic (γ-Linolenic) [13]	C 18:3 *n*-6	184.20	384.00	208.50	96.30
Eicosanoic (arachidic) [14]	C 20:0	0.58	NA	NA	NA
Saturated FA (SFA) [14]		1565.70	1824.00	1751.40	1669.20
Monounsaturated FA (MUFA) [14]		1228.00	2144.00	2543.70	2503.80
Polyunsaturated FA (PUFA) [14]		337.70	480.00	375.30	256.80

NA, not available.

**Table 4 foods-08-00096-t004:** Mineral and vitamin contents of different durian varieties.

Durian Variety	Thailand Variety				Malaysian Variety	Unknown Variety [31]	Unknown Variety [32]	Unknown Variety [33]
	*Monthong*	*Chanee*	*Kradum*	*Kobkatam*	Unknown [15]			
Macrominerals (mg per 100 g fresh weight)								
Calcium [13,14,31,32,33]	4.298–6.134	5.44	3.75	3.21	NA	6.00	40.00	7.00
Phosphorus [13,14,31,32,33]	25.79–33.59	32.96	36.70	37.56	NA	39.00	44.00	44.00
Sodium [13,14,31,32,33]	6.14–15.66	11.84	19.60	21.51	NA	2.00	40.00	1.00
Potassium [13,14,31,32,33]	377.00–489.42	539.20	439.52	438.17	NA	436.00	70.00	601.00
Magnesium [13,14,31,32,33]	19.28–24.87	23.36	23.35	22.79	NA	30.00	NA	NA
Microminerals (mg per 100 g fresh weight)								
Iron [13,14,31,32,33]	0.18–0.23	0.45	0.33	0.36	NA	0.43	1.90	1.30
Copper [13,14,31,32,33]	0.13–0.15	0.27	0.23	0.17	NA	NA	NA	0.12
Manganese [14]	0.23–0.26	NA	NA	NA	NA	NA	NA	NA
Zinc [13,14,31,33]	0.15–0.21	0.45	0.37	0.32	NA	0.28	NA	0.30
Vitamins (μg per 100 g fresh weight)								
A (RAE)	NA	NA	NA	NA	NA	2.00	NA	NA
B_1_/Thiamine	NA	NA	NA	NA	NA	374.00	100.00	100.00
B_2_/Riboflavin	NA	NA	NA	NA	NA	200.00	100.00	100.00
B_3_/Niacin	NA	NA	NA	NA	NA	1074.00	NA	13650.00
B_6_/Pyridoxine	NA	NA	NA	NA	NA	316.00	NA	NA
E/Tocopherol or Tocotrienol (μg per 100 g fresh weight)								
α-tocopherol	NA	NA	NA	NA	3774.00	NA	NA	NA
γ-tocopherol	NA	NA	NA	NA	1013.00	NA	NA	NA
δ-tocopherol	NA	NA	NA	NA	11.00	NA	NA	NA
δ-tocotrienol	NA	NA	NA	NA	1.00	NA	NA	NA

NA, not available; RAE, retinol activity equivalent.

**Table 5 foods-08-00096-t005:** Soluble, insoluble, and total dietary fibre in different durian variety (g per 100 g fresh weight).

Type of Fibre	Soluble [10,12,16]	Insoluble [10,12,16]	Total Dietary Fibre [10,11,12,13,16,31,32,33]
Thailand Variety			
*Monthong*	0.40–1.40	0.80–1.92	1.20–3.39
*Chanee*	1.14	2.44	2.91–3.58
*Kradum*	0.77	1.64	2.41–3.17
*Kan Yao*	1.01	0.60	1.61
*Puang Manee*	0.74	1.95	2.69
*Kobtakam*	NA	NA	2.41
Unknown variety	NA	NA	3.80
Unknown variety	NA	NA	0.90
Unknown variety	NA	NA	3.50

NA, not available.

**Table 6 foods-08-00096-t006:** Bioactive compounds of different durian varieties (mg/μg per 100 g fresh weight).

Bioactive Compounds	Durian Variety											
	Malaysian Variety					Thailand Variety						Unknown Variety [23]
	*Chaer Phoy*	*Yah Kang*	*Ang Jin*	D11	Unknown	*Chanee*	*Kan Yao*	*Puang Manee*	*Kradum*	*Monthong*	*Kobtakam*	
**Polyphenols**												
Total polyphenols [10,11,12,13,16,17,18,19,20,21,22]	67.12 mg GAE	80.45 mg GAE	97.78 mg GAE	71.13 mg GAE	99.00 mg GAE	21.44–321.20 mg GAE	283.30 mg GAE	310.50 mg GAE	94.18–271.50 mg GAE	56.18–374.30 mg GAE	94.18 mg GAE	79.15 mg GAE
**Flavonoids**												
Total flavonoids [10,12,18,29,32,34,35,36,37,38,39]	22.56 mg CE	22.22 mg CE	22.50 mg CE	20.58 mg CE	NA	1.90–81.60 mg CE	3.51–72.10 mg CE	3.24–18.10 mg CE	4.48–19.80 mg CE	4.49–93.90 mg CE	NA	NA
Flavanone												
Hesperetin [16]	NA	NA	NA	NA	NA	321.15 μg	260.99 μg	640.79 μg	1110.23 μg	562.98 μg	NA	NA
Hesperidin [19]	NA	NA	NA	NA	NA	NA	NA	NA	NA	200.00 μg	NA	NA
Flavonol												
Quercetin [18,19,20]	NA	NA	NA	NA	NA	2.22 mg	2.44 mg	2.18 mg	NA	1.20–2549.30 mg	NA	NA
Morin [19]	NA	NA	NA	NA	NA	NA	NA	NA	NA	110.00–550.00 μg	NA	NA
Rutin [18]	NA	NA	NA	NA	NA	492.41 μg	NA	733.20 μg	163.90 μg	912.05 μg	NA	NA
Kaempferol [16,19]	NA	NA	NA	NA	NA	479.09 μg	644.80 μg	430.18 μg	131.64 μg	830.26–2200.00 μg	NA	1310.00 mg
Myricetin [19]	NA	NA	NA	NA	NA	NA	1559.56 μg	964.47 μg	2159.27 μg	320.00–2087.83 μg	NA	1010.00 mg
***Bioactive Compounds***	**Durian Variety**											
	**Malaysian Variety**					**Thailand Variety**						**Unknown Variety [23]**
	***Chaer Phoy***	***Yah Kang***	***Ang Jin***	**D11**		***Chanee***	***Kan Yao***	***Puang Manee***	***Kradum***	***Monthong***	***Kobtakam***	
**Flavonoids**												
Flavone												
Luteolin [21]	NA	NA	NA	NA		364.92 μg	279.29 μg	509.09 μg	287.69 μg	338.22 μg	NA	NA
Apigenin [21]	NA	NA	NA	NA		739.42	763.83 μg	509.09 μg	791.94 μg	620.00–665.89 μg	NA	NA
Total flavanols [11,12,14,17,18,19,20]	NA	NA	NA	NA		0.15 mg CE	0.13 mg CE	0.15 mg CE	0.13 mg CE	0.18 mg CGE–5.18 mg CE	NA	NA
Total anthocyanins [15,17,38]	NA	NA	NA	NA		0.38 mg CGE	0.34 mg CGE	0.37 mg CGE	0.32 mg CGE	0.39–633.44mg CGE	NA	NA
**Phenolic Acids**												
Cinnamic acid [19]	NA	NA	NA	NA		NA	NA	NA	NA	600.00–660.00 μg	NA	1510.00 mg
Caffeic acid [19,21]	NA	NA	NA	NA		NA	NA	NA	NA	31.08–490.00 μg	NA	NA
*p*-Coumaric acid [19,21]	NA	NA	NA	NA		NA	NA	NA	NA	29.22-600.00 μg	NA	NA
Ferulic acid [18,21]	NA	NA	NA	NA		215.95 μg	NA	158.67 μg	NA	414.40 μg	NA	NA
*p*-Anisic acid [22]	NA	NA	NA	NA		NA	NA	NA	NA	1.48 μg	NA	NA
Gallic acid [18]	NA	NA	NA	NA		1416.00 μg	NA	4760.10 μg	NA	2072.00 μg	NA	NA
Vanillic acid [19,22]	NA	NA	NA	NA		NA	NA	NA	NA	20.72–300.00 μg	NA	NA
***Bioactive Compounds***	**Durian Variety**											
	**Malaysian Variety**					**Thailand Variety**						
	***Chaer Phoy***	***Yah Kang***	***Ang Jin***	**D11**	**Unknown [15]**	***Chanee***	***Kan Yao***	***Puang Manee***	***Kradum***	***Monthong***	***Kobtakam***	
**Carotenoids**												
Total carotenoids [11,17,24]	7.10 μg BCE	5.13 μg BCE	6.02 μg BCE	8.22 μg BCE	NA	4400.00–6000.00 μg	NA	NA	NA	222.88–1167.00 μg	NA	
β-Carotene [13,20,21,24,25]	NA	NA	NA	NA	201.00 μg	84.54–4429.00 μg	54.17 μg	320.87 μg	232.44–250.20 μg	35.92–4250.00 μg	385.84 μg	
α-Carotene [13,20,21,24,25]	NA	NA	NA	NA	37.00 μg	47.23–1329.00 μg	8.61 μg	38.55 μg	52.54–79.09 μg	7.79–343.00 μg	263.54 μg	
β-Crptoxanthin [13,16]	NA	NA	NA	NA	7.00 μg	17.58 μg	4.87 μg	17.80 μg	26.80 μg	5.85 μg	ND/NA	
Lycopene [13,16]	NA	NA	NA	NA	12.00 μg	11.62 μg	1.38 μg	17.47 μg	6.91 μg	2.80 μg	ND/NA	
Lutein [13,16,24,25]	NA	NA	NA	NA	11.00 μg	14.00–41.28 μg	7.21 μg	18.16 μg	32.35–54.21 μg	7.96–41.75 μg	72.23 μg	
Zeaxanthin [13,16,24,25]	NA	NA	NA	NA	37.00 μg	0.09–37.47 μg	11.37 μg	20.21 μg	49.44 μg	0.14–11.95 μg	ND/NA	
Tannins [16,18,21,22]	NA	NA	NA	NA	NA	NA	NA	NA	NA	29.60–296.00 μg	NA	
Ascorbic acid [11,14,17,20,21,22]	2.41 mg	2.21 mg	1.93 mg	2.56 mg	25.18 mg	NA	NA	NA	NA	54.76–347.80 mg	NA	

NA, not available; GAE, gallic acid equivalent; CE, catechin equivalent; CGE, cyanidin-3-glucoside equivalent; ND, not detected; BCE, β-carotene equivalent.

**Table 7 foods-08-00096-t007:** Antioxidant activities of different durian varieties (μM Trolox equivalents per 100 g fresh weight).

Type of Antioxidant Activity Assay	DPPH [12,13,14,22,40,41]	FRAP [12,13,14,18,22]	ORAC[13]	CUPRAC[12,14,18,22]	ABTS [12,14,18,22,41]	H-ORAC[15]
Thailand Variety						
*Monthong*	97.93–1366.15	71.84–749.08	1903.40	427.65–1075.60	265.86–2352.70	NA
*Chanee*	128.00–245.60	232.10–457.43	2304.00	955.40	2091.40	NA
*Kradum*	250.20	667.20	2793.90	806.50	1773.20	NA
*Kan Yao*	209.09	204.70	NA	845.50	1843.60	NA
*Puang Manee*	NA	244.90	NA	924.90	2020.40	NA
*Kobtakam*	192.60	513.60	2343.30	NA	NA	NA
Malaysian Variety						
Unknown	NA	NA	NA	NA	NA	1838.00
Unknown Variety						
Unknown [23]	NA	NA	NA	NA	498.00	NA

NA, not available; DPPH, 1,1-diphenyl-2-picrylhydrazyl radical; FRAP, ferric ion reducing antioxidant power; ORAC, oxygen radical absorbance capacity; CUPRAC, cupric reducing antioxidant capacity; H-ORAC, hydrophilic oxygen radical absorbance capacity; ABTS, 2,2’-azino-bis-3-ethylbenzthiazoline-6-sulphonic acid.

**Table 8 foods-08-00096-t008:** Mean relative amounts of volatiles identified in different durian varieties.

Compounds	Relative Amount in ng per g fresh weight						
	Malaysian Variety			Indonesian Variety			
	D101	D2	D24	*Hejo*	*Matahari*	*Ajimah*	*Sukarno*
Sulphur compounds							
Ethanethiol [7,42]	5480.00	4260.00	3550.00	ND	5.40	50.70	36.40
Propanethiol [7,42]	5000.00	2720.00	2720.00	ND	18.00	31.10	ND
Methyl ethyl disulphide [42]	NA	NA	NA	ND	ND	ND	11.50
Diethyl disulphide [7,42]	12420.00	1585.00	18760.00	24.40	323.90	245.20	188.40
Ethyl propyl disulphide [7,42]	3630.00	3350.00	9040.00	2.30	43.20	11.30	4.60
Bis(ethylthio)methane [42]	NA	NA	NA	49.30	105.40	246.10	118.20
Diethyl trisulphide [7,42]	5970.00	14680.00	2520.00	10.20	185.50	213.60	72.30
3,5-Dimethyl-1,2,4- trithiolane (isomer 1) [7,42]	470.00	1460.00	1740.00	1.50	10.60	20.80	2.00
3,5-Dimethyl-1,2,4- trithiolane (isomer 2) [7,42]	590.00	1470.00	1710.00	1.00	10.60	17.70	1.10
1,1-Bis(methylthio)- ethane [7]	NA	NA	NA	5.30	14.50	5.80	3.00
1,1-Bis(ethylthio)-ethane [7,42]	420.00	490.00	710.00	1.90	15.80	5.90	10.20
3-Mercapto-2- methylpropanol[7]	NA	NA	NA	2.50	21.90	2.80	4.70
Dipropyl trisulphide [7]	120.00	160.00	110.00	NA	NA	NA	NA
Dipropyl disulphide [7]	200.00	110.00	1030.00	NA	NA	NA	NA
1-(ethylthio)-1-(methylthio)-Ethane [7]	660.00	140.00	660.00	NA	NA	NA	NA
S-propyl ethanethioate [7]	340.00	60.00	320.00	NA	NA	NA	NA
S-ethyl ethanethioate [7]	90.00	ND	310.00	NA	NA	NA	NA
1-(methylthio)-propane [7]	270.00	ND	130.00	NA	NA	NA	NA
Total	35660.00	30485.00	43310.00	98.40	754.80	851.00	452.40
Alcohols							
Ethanol [7,42]	720.00	1090.00	590.00	419.90	688.80	843.40	1091.30
2-Methyl-1-butanol [42]	NA	NA	NA	ND	ND	17.40	ND
3-Methyl-1-butanol [42]	NA	NA	NA	10.50	ND	ND	14.60
2,3-Butanediol [42]	NA	NA	NA	4.60	ND	ND	11.70
Total	720.00	1090.00	590.00	435.00	688.80	860.80	1117.60
Ketones							
3-Hydroxy-2-butanone [42]	NA	NA	NA	56.20	84.20	71.30	64.10
Aldehydes							
Acetaldehyde [42]	NA	NA	NA	44.90	62.20	33.90	ND
Esters							
Ethyl acetate [7,42]	280.00	610.00	930.00	28.10	52.40	34.80	31.20
Methyl propanoate [7,42]	970.00	880.00	700.00	16.40	52.20	ND	ND
Ethyl propanoate [7,42]	3110.00	1850.00	2530.00	386.60	719.50	742.30	0.00
Methyl-2-methylbutanoate [7,42]	4070.00	2330.00	2290.00	86.20	105.60	85.40	24.90
Ethyl butanoate [7,42]	850.00	2220.00	40.00	73.20	131.90	252.30	83.20
Propyl propanoate [7,42]	4630.00	1740.00	3810.00	ND	88.40	ND	ND
Ethyl 2-methylbutanoate [7,42]	460.00	510.00	500.00	2938.40	2373.40	3846.70	1085.90
Diethyl carbonate [42]	NA	NA	NA	ND	9.70	ND	7.40
Propyl 2-methylbutanoate [7,42]	126.70	4770.00	113.00	109.30	208.50	192.0	11.80
Propyl butanoate [7,42]	950.00	630.00	950.00	3.50	16.30	ND	ND
Propyl 3-methylbutanoate [7,42]	19.00	ND	380.00	237.90	ND	ND	ND
Ethyl 2-butenoate [7,42]	ND	140.00	ND	ND	252.20	397.60	132.10
Methyl hexanoate [7]	320.00	1700.00	ND	ND	ND	ND	ND
Ethyl (2E)-2-pentenoate [42]	NA	NA	NA	4.50	ND	ND	ND
Ethyl 3-hexanoate [42]	NA	NA	NA	ND	ND	93.02	32.30
Propyl hexanoate [7,42]	580.00	500.00	310.00	3.10	24.20	3.90	ND
Propyl tiglate [42]	NA	NA	NA	12.80	ND	ND	ND
Ethyl heptanoate [7,42]	150.00	250.00	150.00	42.40	111.20	74.80	ND
Methyl octanoate [7,42]	220.00	100.00	ND	4.30	26.90	ND	ND
Ethyl octanoate [7,42]	550.00	550.00	260.00	91.0	174.60	108.40	45.90
Ethyl (4Z)-4-octenoate [42]	NA	NA	NA	ND	17.40	ND	ND
Ethyl-2,4-hexadienoate [42]	NA	NA	NA	ND	ND	3.10	2.60
Ethyl-3-hydroxybutanoate [42]	NA	NA	NA	6.10	12.20	23.40	16.80
Propyl octanoate [42]	NA	NA	NA	ND	14.50	ND	ND
Ethyl-2-octenoate [42]	NA	NA	NA	2.20	ND	ND	ND
Ethyl decanoate [42]	NA	NA	NA	10.90	8.20	10.40	11.1
Ethyl 2-methylpropanoate [7]	460.00	510.00	520.00	NA	NA	NA	NA
Propyl acetate [7]	190.00	90.00	560.00	NA	NA	NA	NA
Methyl butanoate [7]	300.00	450.00	ND	NA	NA	NA	NA
Ethyl 3-methylbutanoate [7]	190.00	220.00	220.00	NA	NA	NA	NA
3-methylbutyl propanoate [7]	730.00	ND	600.00	NA	NA	NA	NA
Total	19155.70	20050.00	14863.00	3947.60	4399.30	5868.12	1429.10

NA, not available; ND, not detected.

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
