# Peer review of "Bioactive Compounds, Nutritional Value, and Potential Health Benefits of Indigenous Durian (Durio Zibethinus Murr.): A Review"

_foods, 2019, doi:10.3390/foods8030096_

Round 1
Reviewer 1 Report
The manuscript is well-written and can be published in its current form
Author Response
There is no comments or suggestion made by the reviewer. No changes has been made.
Reviewer 2 Report
The review presents information of interest in relation to the nutritional composition and volatile compounds of Durian fruit. However, some essential properties of this fruit have not been referenced in the review. The importance of this fruit is mostly connected with its composition and antioxidant properties. There has been a great interest in the potential health benefits of exotic fruits due to their antioxidant content and bioactive compounds. In the literature there are several reports available on determining the antioxidant capacity of Durian fruit, for example: Ratiporn Haruenkit, et al. Food Chem. 118 (2010) 540-547; Shela Gorinstein et al. Food Res. Int. 44 (2011) 2222-2232; Lee-Hoon, et al. Food Chem 168 (2015) 80-89.
On the other hand, the health properties of Durian are based also on fatty acid composition (polyunsaturated fatty acids). Durian is rich in n-3 fatty acids, compared to some other fruits and this information has not been emphasized adequately.
-Nutritional Composition of Different Durian Varieties. Line 78: it would be necessary to include and comment some references regarding the composition in unsaturated fatty acids.
-Include a new section on antioxidant capacity and comment on the different methods used for its determination. This section should also be included in the Conclusions Section.
Author Response
Response to Reviewer 2 Comments
We would like to thank you the reviewer for their valuable comments and suggestions. We have made the changes accordingly.
Point 1: However, some essential properties of this fruit have not been referenced in this review. The importance of this is mostly connected with its composition and antioxidant properties.
Point 2: Include a new section on antioxidant capacity and comment on the different methods used for its determination. This section should also be included in the Conclusions Section.
Response 1 and 2: We agreed with the reviewer comment and have added one table (line 190 to 194, in red) and discussion for the antioxidant capacity in line 178 to 189 (in red). The different types of in vitro methods used for antioxidant capacity determination were added in line 178 to 183 (in red). A new information was added in ‘conclusions’ section (line 342 to 344, in red).
Point 3: Durian is rich in n-3 fatty acids, compared to some other fruits and this information has not been emphasized adequately.
Point 4: It would be necessary to include and comment some references regarding the composition in unsaturated fatty acids.
Response 3 and 4: We agreed with the reviewer comments and have added one table and discussion for fatty acid composition (line 78 to 87 and line 98 to 99, in red).

Reviewer 3 Report
The reviewed article is a collection of current knowledge about the chemical composition and impact on human health of tropical fruit - durian.
I have several comments to the manuscript content:
- "in vitro" and "in vivo" please write in italics (lines: 16, 295, 312 and 325)
- there should be full stops after the table titles
- there is a custom to write "polyphenols" not "(poly)phenols
- line 47 - From the order of the sentence, it appears that compounds such as ascorbic acid or carotenoids are polyphenols. The sentence should be rewritten. Suggestion: Durian is also rich in ascorbic acid, carotenoids and polyphenolas such as flavonoids (flavanones, flavonols, flavones, flavanols), phenolic acids and anthocyanins.
- line 69 - delete "and" at the end of the sentence
- abbreviation "NA" should be explained under each table in which it occurs, not only under the first one
- line 127 - add "s" at the end of the word "flavanone" (plural)
- line 129 - add "s" at the end of the word "flavanol" (plural)
- line 131 - morin belongs to group of flavonol, not flavanol. Please correct it.
- line 221 - delete (explanation of abbreviations should be under the table and not above it)
- line 292 - remove the word "flavonoids" and leave only "polyphenols", because flavonoids are also polyphenols
- note the spaces in line 39.
Author Response
Response to Reviewer 3 Comments
We would like to thank you the reviewer for their valuable comments and suggestions. We have made the changes accordingly.
Point 1: “In vitro” and “in vivo” please write in italics (lines: 16, 295, 312 and 325)
Response 1: We agreed with the reviewer comment to italicize the word “in vitro” and “in vivo” throughout the text (in red).
Point 2: There should be full stops after table titles
Response 2: We agreed with the reviewer and have added full stops in all table title (in red).
Point 3: There is a custom to write “polyphenols” not “(poly)phenols
Response 2: We agreed with the reviewer comment to use the term polyphenols throughout the text (in red).
Point 4: Line 47 – From the order of sentence, it appears that compounds such as ascorbic acid or carotenoids are polyphenols. The sentence should be rewritten.
Response 4: We agreed with the reviewer comment to rewrite the sentence (line 48-49).
Point 5: Line 69 – delete “and” at the end of the sentence
Response 5: We agreed with the reviewer comment and has deleted the word “and” in line 69.
Point 6: Abbreviation “NA” should be explained under each table in which occurs, not only under the first one
Response 6: We agreed with the reviewer to explain the word “NA” under each table. Changes were made in all tables (in red).
Point 7: Line 127 – add “s” at the end of the word “flavanone” (plural)
Response 7: We agreed with the reviewer comment and have added the letter “s” (line 133).
Point 8: Line 129 – add “s” at the end of the word “flavanol” (plural)
Response 8: We agreed with the reviewer comment and have added the letter “s” (line 134).
Point 9: Line 131 – Morin belongs to group of flavonol, not flavanol. Please correct it.
Response 9: We agreed with the reviewer comment and have changed the ‘flavonol’ to flavanol (line 137).
Point 10: Line 221 – Delete (explanation of abbreviations should be under the table and not above it)
Response 10: We already moved the explanation under the table as suggested (line 167).
Point 11: Line 292 – remove the word “flavonoids” and leave only “polyphenols”, because flavonoids are also polyphenols.
Response 11: We agreed with the reviewer comment to remove the word “flavonoids” (line 314).
Point 12: Note the spaces on line 39
Response 12: We agreed with the reviewer comment to delete the ‘spaces’ (line 40).

Round 2
Reviewer 2 Report
Now more useful information is available regarding the nutritional composition of the Durian.
Author Response
Comment from reviewer 2:
Now more useful information is available regarding the nutritional composition of the Durian.
Response: We would like to thank you the reviewer for their feedback.